# Fermentative Production of Diacylglycerol by Endophytic Fungi Screened from *Taxus chinensis var. mairei*

**DOI:** 10.3390/foods12020399

**Published:** 2023-01-14

**Authors:** Wenqiang Xu, Haoran Bi, Hong Peng, Ling Yang, Hongwei He, Guiming Fu, Yuhuan Liu, Yin Wan

**Affiliations:** 1State Key Laboratory of Food Science and Technology, Nanchang University, Nanchang 330047, China; 2Engineering Research Center of Biomass Conversion, Ministry of Education, Nanchang University, Nanchang 330047, China

**Keywords:** diacylglycerol, endophytic fungi, screening, fermentation, fatty acid composition

## Abstract

Diacylglycerol (DAG) production by microbial fermentation has broad development prospects. In the present study, five endophytic fungi which could accumulate DAG were screened from *Taxus chinensis var. mairei* by using potato dextrose agar plate and flask cultivation in potato dextrose broth culture medium. The strains were biologically identified based on morphological features and semi-quantitative PCR. The identification results indicated that the five strains belonged to different genera: *Fusarium annulatum* (*F. annulatum,* coded as MLP41), *Trichoderma dorotheae* (*T. dorotheae,* coded as MLG23), *Colletotrichum aeschynomenes* (*C. aeschynomenes,* coded as MLY23), *Pestalotiopsis scoparia* (*P. scoparia,* coded as MLY31W), and *Penicillium cataractarum* (*P. cataractarum,* coded as MLGP11). The crude lipids from the strains and their corresponding triacylglycerol, 1,2-DAG, and 1,3-DAG fractions separated via thin-layer chromatography were mainly composed of palmitic acid, stearic acid, oleic acid, and linoleic acid, which in total accounted for higher than 94% of the content. The effects of fermentation conditions on the DAG productivity were discussed, and the yields of DAG were determined based on the ^1^H NMR spectra of crude lipids. The highest total DAG yields of *F. annulatum*, *T. dorotheae*, *C. aeschynomenes*, *P. scoparia*, and *P. cataractarum* were 112.28, 126.42, 189.87, 105.61, and 135.56 mg/L, respectively. *C. aeschynomenes* had the strongest potential to produce DAG. The results showed that this may be a new promising route for the production of DAG via fermentation by specific endophytic fungi, such as *C. aeschynomenes*.

## 1. Introduction

With the improvement of living standards, people’s intake of high-fat foods has increased, resulting in an increase in the proportion of people suffering from overweight, obesity, hypertension, hyperlipidemia, and other diseases year by year. According to a World Health Organization survey report, worldwide, 39% of 18-year-old adults were overweight and 13% were obese in 2016. Therefore, the development of lipids that are beneficial to human health is of great significance for reducing cardiovascular and cerebrovascular diseases, obesity, and the like.

As an important structural lipid, diacylglycerol (DAG) has nearly similar properties to regular dietary fats, but is lower in calories than triacylglycerol (TAG). DAG does not spoil the texture of the products and sensory attributes when it is added to food as a fat substitute [1]. DAG has been evaluated by the U.S. Food and Drug Administration (FDA) as a GRAS (Generally Recognized as Safe) substance, and is currently used as an additive in the food, cosmetic, and pharmaceutical industries to play a role as non-ionic emulsifiers and stabilizers [1]. Studies have shown that DAG, as an anti-obesity agent, can reduce postprandial fatty acid levels [2], reduce TAG levels and LDL cholesterol in the serum and liver, and inhibit fat accumulation, thereby having a positive impact on human health [1,3]. According to the reports in literature, DAG also shows potential to promote cardiovascular health and play a major role in tumor immunosurveillance [3,4]. Given the important physiological functions of DAG, it is of great significance to replace TAG with DAG as the main component of edible oils and fats.

Though DAG is one of the components in common natural oils and fats, its content is very low, ranging from 1% to 5% and generally no more than 10% [1,5,6]. It is unrealistic to separate DAG directly from common natural oils and fats. Therefore, exploring technologies for DAG production has received extensive attention in the functional food industry. In general, there are two approaches to produce DAG. One is the hydrolysis or glycerolysis of TAG via chemical or enzymatic methods [1,7,8,9]. Alternatively, fatty acids (FAs) are esterified with glycerol to produce DAG [1,9]. However, the traditional DAG production methods, including chemical and enzymatic methods, have some disadvantages, such as high energy consumption and high contents of by-products using chemical methods [1,7,9], high costs, low reaction rates, and poor stability using enzymatic methods [1,8,9]. In addition, it is difficult to control the quality of DAG products [1,7,9]. All of these shortcomings inhibit the efficient production of DAG to a certain extent.

An oleaginous microorganism is possibly another important source of DAG. Obtaining DAG via oleaginous microbial fermentation has many advantages, such as less land occupation, a short cultivation period, and similar fatty acid (FA) composition to vegetable oils [10]. Among these different oleaginous microorganisms, including yeast, algae, bacteria, and fungi, filamentous fungi are easier harvest, especially when they are grown as pellets or mycelia [11]. So far, there are few reports on the use of microbial fermentation to produce DAG [12,13]. Vulevic and Gibso [12] reported that a human gut isolate of *Bifidobacterium longum* biovar *infantis* could synthesize 1,2-sn-DAG in vitro using anaerobic static batch-culture fermentation in the presence of phosphatidylcholine, but its yield was very low, only 84.97 nmol/mL. The key step of DAG production via fermentation is to obtain strains with high DAG production. Almost each plant on earth is the host of one or more endophytic fungi. Because of co-evolution, horizontal gene transfer occurs between the plants and their endophytic fungi, resulting in endophytic fungi being capable of producing the same metabolites as their plant hosts [14,15]. Considering this, the endophytic fungi isolated from oil plants are expected to have DAG-producing ability. Previous studies have shown that *Taxus* species (*Taxus* sp.) are a good source of lipids and some special fatty acids [16,17,18]. Thus, the endophytic fungi of *Taxus* sp. are associated with the great possibility of acquiring special lipids via accumulation, such as high DAG-containing lipids.

In the present study, endophytic fungal strains with lipid-production ability, with DAG as one of the main components, were isolated and screened from *Taxus chinensis var. mairei* (*T. chinensis*). Then, the screened isolates were biologically identified. The FA composition of the lipids extracted from these screened endophytic fungi was analyzed. Finally, the effects of fermentation conditions on DAG productivity were analyzed. The goal of the present study was to obtain one or more endophytic fungal strains with great potential for DAG production.

## 2. Materials and Methods

### 2.1. Source of Plant Tissues

Healthy and fresh leaves, stems, roots, and barks were collected from *T. chinensis*, which was growing in Meiling National Forest Park in Nanchang City, China. After collection, the samples were immediately put in sterile zip-lock bags and transported carefully to the laboratory; then, they were stored at 4 °C for further use.

### 2.2. Isolation of Endophytic Fungi

The isolation of endophytic fungi from the plant tissues was carried out according to Ismaiel et al. [19]. First, the roots and stems of *T. chinensis* were scraped to remove the epidermis before being cut into 4–5 cm-long sections, and the leaves were cut into pieces with a length of 1–2 cm. The surfaces of all cut plant tissues were rinsed with sterile distilled water 2–3 times and then rinsed further with 70% ethanol for 30 s. After that, the tissues were soaked in 5% sodium hypochlorite for 30 s, followed by rinsing 2 times with sterile distilled water. After being dried on the sterile dried filter paper, the tissues were cut into small pieces (about 5 mm × 5 mm) and inoculated onto potato dextrose agar (PDA) plates containing 10 mg/L streptomycin and 3 mg/L tetracycline to inhibit bacterial growth. The plates were incubated at 28 °C in dark and checked every 24 h. It was found that the mycelium could grow out within 3–7 days. After that, according to the color, shape, and secretion of the grown colonies, the tip hyphae at the edges of the colonies with different appearances were immediately picked up with an inoculation loop and transferred onto fresh PDA plates. The inoculated plates were incubated again at 28 °C for 3–7 days in dark. This repetitive re-plating operation was repeated until each colony was a pure culture. The pure strains were coded separately and preserved using the test tube slant method for further use.

### 2.3. Screening of DAG-Producing Endophytic Fungal Strains

#### 2.3.1. Flask Cultivation

Two pieces of mycelium with a diameter of 1 cm were taken out from the PDA plate containing the evenly distributed pure endophytic fungus strain after incubation for 3–7 days and introduced into 250 mL flasks containing 100 mL potato dextrose broth (PDB) culture medium. The PDB culture medium contained 26.0 g/L potato dextrose broth, 0.5 g/L peptone, 0.8 g/L yeast extract, 0.5 g/L MgSO_4_·7H_2_O, 3.0 g/L (NH_4_)_2_SO_4_, and 2.0 g/L KH_2_PO_4_. After cultivation at 28 °C for 7 days with a shaking speed of 120 rpm, the fermented broth and fungal biomass was separated via filtration using four layers of Nylon gauze with a 100 mesh. The fresh mycelium was washed with distilled water to remove the culture medium contaminant on the surface of the mycelium and then freeze-dried for 48 h to obtain the fungal biomass. The productivity of cell dry weight (CDW) was determined according to formula (1).
(1)$$\mathrm{CDW}\;(g/L)=\frac{m_{1}}{V}$$
where *m*_1_ was the weight of the obtained dry fungal biomass (g) and *V* was the volume of fermentation medium (L).

#### 2.3.2. Screening of Lipid-Producing Endophytic Fungal Strains

After being ground fully into powder, the lipids were extracted from the dry fungal biomass powder using petroleum ether as an extractant in Soxhlet extractors. After extraction for 8 h, the extracts were then concentrated to dryness through rotary evaporation to obtain lipids. The lipid yield was calculated according to formula (2). Finally, the strains with strong lipid-producing ability were screened out according to the lipid contents in the mycelium.
(2)$$\mathrm{Lipid}\;\mathrm{yield}\;(g/L)=\frac{m_{2}}{m_{3}}\times m_{4}$$
where *m*_2_ was the productivity of CDW (g/L), *m*_3_ was the mass of the mycelium powder weighed for lipid extraction (g), and *m*_4_ was the mass of the extracted lipids (g).

#### 2.3.3. Screening of DAG-Producing Endophytic Fungal Strains and Analysis of FA Composition

The extracted lipids were dissolved separately in n-hexane to obtain solutions with a lipid concentration of 3% (g/mL). Then, the solutions were spotted on G silica gel plates (10 cm × 20 cm) for thin-layer chromatography (TLC). The developing solvent was composed of n-hexane, ethyl acetate, and acetic acid (70:30:0.1, mL/mL/mL). The silica plates were visualized under a UV lamp after spraying with 0.2% 2,7-dichlorofluoresce in ethanol solution. Based on the TLC results, DAG-producing endophytic fungal strains could be screened.

The silica bands of TAG, 1,3-DAG, 1,2-DAG, and monoacylglycerol (MAG) were scraped off separately and extracted with a small amount of n-hexane to obtain lipid components for FA composition analysis. Before FA composition analysis, the extracts were converted into their fatty acid methyl esters (FAMEs) referring to the method in the literature with slight modifications [20]. Briefly, 1 mL of the solvent mixture containing the lipid extracts was put into a tube and evaporated at room temperature to remove n-hexane. Then, 5 mL of a 2 mol/L methanol solution of sodium methoxide was added into the solvent-evaporated lipids and mixed completely with a vortex for 10 min at 50 °C to carry out the *trans*-esterification reaction. After that, 2 mL of n-hexane was added into the mixture to extract the FAME fraction. Finally, the upper n-hexane layers were analyzed with gas chromatography-mass spectrometry (GC-MS) after being dehydrated with a proper amount of anhydrous Na_2_SO_4_. The analysis was performed on an Agilent 7890B-7000D GC-MS equipped with an Agilent HP-PONA capillary column (50 m × 0.2 mm × 0.5 μm). The mass signals were obtained in the range of 50–550 m/z. The ionization source EI had an electron energy of 70 eV with an ion source temperature of 230 °C. The carrier gas of helium was maintained at a flow rate of 1.0 mL/min. The injection volume was 1 μL with a split ratio of 50:1. Before injection, the samples were filtered with a 0.22 μm membrane. The following temperature program was adopted: held initial temperature of 80 °C for 5 min, then raised to 290 °C with a heating rate of 4 °C/min, and finally held at 290 °C for 5 min. The acquired mass spectral information of FAMEs was searched using the NIST14.L spectral library, and the area normalization method was used for quantitative analysis.

At the same time, the morphological features of DAG-producing endophytic fungal strains were observed with a UB103i upright biomicroscope (Chongqing UOP Co., Ltd., Chongqing, China) after the mycelium was fixed on the glass slide.

### 2.4. Biological Identification of DAG-Producing Endophytic Fungal Strains

#### 2.4.1. DNA Extraction and Semi-Quantitative PCR

The fungal isolates cultured in PDA medium at 28 °C for 4–6 days were harvested, and their genomic DNA was then extracted using the Tsingke Plant DNA Extraction Kit (Beijing Tsingke Biotechnology Co., Ltd., Beijing, China) according to the manufacturer’s instruction. For PCR amplification, two kinds of universal primers, ITS1 (TCCGTAGGTGAACCTGCGG) and ITS4 (TCCTCCGCTTATTGATATGC), were used. ITS regions were amplified from the genomic DNA via PCR according to the manufacturer’s protocol (Beijing Tsingke Biotechnology Co., Ltd., Beijing, China). The PCR reaction mixture was composed of 1 μL fungal DNA, 45 μL Tsingke TSE101 Ready-to-use Rapid PCR Premix, 2 μL ITS1, and 2 μL ITS4 with a final volume of 50 μL of the reaction mixture. PCR was performed in a 2070 Thermal Cycler (Applied Biosystems, USA) with an initial denaturation step at 98 °C for 2 min followed by 35 cycles of 98 °C for 10 s, 56 °C for 10 s, and 72 °C for 10 s/kb, and a final extension step at 72 °C for 5 min. The amplified PCR products were subjected to agarose gel electrophoresis and visualized under a UV-trans-illuminator following bromophenol blue staining. Then, the prepared PCR products were sequenced with a 3700 Genetic Analyzer (Applied Biosystems, Waltham, MA, USA). The final obtained sequences were submitted to the National Centre of Biological Information (NCBI) database.

#### 2.4.2. Phylogenetic Analysis

The sequencing results were spliced using Contig Express, and the inaccurate parts at both ends were removed. The obtained sequences were then analyzed using the BLAST algorithm, and closely related phylogenetic sequences were obtained from the NCBI database. The phylogenetic tree was constructed using the phylogenetic analysis software MEGA 11 based on the Neighbour-joining (NJ) method using Kimura two-parameter distances. The quality of the branching patterns for NJ was assessed through bootstrap resampling of the datasets with 1000 replications.

### 2.5. Evaluation of Effect of Culture Conditions on DAG Productivity

The culture conditions of the screened DAG-producing endophytic fungi that resulted in relatively high DAG productivity were optimized. The five key parameters influencing DAG productivity in endophytic fungal cells were taken into account, including the cultivation time, inoculation dosage, temperature, rotating speed of flasks (dissolved oxygen), and the mass ratio of carbon to nitrogen (C/N ratio).

For estimating the influencing factors, including the cultivation time, inoculation dosage, temperature, and rotating speed of flasks, the endophytic fungal isolates were cultured in sterile fermentation K&R (Kendrick and Ratledge) medium. The K&R medium is rich in carbon but poor in nitrogen, which can promote lipid accumulation [21]. Since some of the screened strains did not produce spores, the spore suspension could not be prepared. Thus, the inoculations were carried out in the form of inoculated mycelial blocks with a diameter of 1 cm after culture at 28 °C for 7 days on PDA plates, and one mycelial block per 100 mL of medium was recorded as 1/100. The set mycelial blocks of each endophytic fungal strain were inoculated into 400 mL of sterile fermentation medium in 1000 mL baffled flasks and incubated at the set temperature for the set time with the set shaking speed. The composition of the fermentation K&R medium in g/L was glucose (76), yeast extract (1.5), (NH_4_)_2_SO_4_ (2.3), MgSO_4_·7H_2_O (1.5), KH_2_PO_4_ (7.0), Na_2_HPO_4_ (2.0), CaCl_2_ (0.075), FeCl_3_·6H_2_O (0.008), ZnSO_4_·7H_2_O (0.001), CuSO_4_·5H_2_O (0.0001), CoSO_4_·7H_2_O (0.0001), and MnSO_4_·5H_2_O (0.0001). For estimating the influence of the C/N ratio (g/g), the addition of glucose was fixed as 76 g/L, and the addition of the nitrogen source was adjusted according to the set C/N ratio. After fermentation was finished, the fermentation broth was filtered with four layers of Nylon gauze with a 100 mesh to obtain fungal biomass and fermented broth. Then, the yields of lipids were determined according to formula (2).

### 2.6. ^1^H NMR Analysis of the Extracted Lipids and Determination of DAG Yield

Before ^1^H NMR analysis, 10 mg of the extracted lipids was dissolved in 425 μL deuterated chloroform (CDCl_3_), which contained tetramethylsilane (TMS) as an internal reference. The mixture was introduced into an NMR tube with a diameter of 5 mm. The ^1^H NMR spectra were recorded on a Bruker Avance 600 spectrometer operating at 600 MHz. The acquisition parameters were as follows: spectral width 10,822 Hz; number of scans, 64; relaxation delay, 3 s; acquisition time, 3 s; and pulse width, 90°. The ^1^H NMR spectrum of the mixture of standard compounds (TAG, 1,2-DAG, 1,3-DAG, and MAG) was also recorded.

## 3. Results and Discussion

### 3.1. Isolation and Screening of DAG-Producing Endophytic Fungi

As shown in Appendix A, 38 total endophytic fungal strains were isolated from the plant tissues of *T. chinensis*. It can be seen from Appendix A that the endophytic fungi in *T. chinensis* showed rich species diversity, and the numbers of endophytic fungal strains were also quite different in different parts of the same tree species. The numbers of endophytic fungal strains isolated from the roots, leaves, barks, and stems were 17, 9, 7, and 5, accounting for 44.74%, 23.68%, 18.42%, and 13.16%, respectively. The proportion of endophytic fungi in the roots was higher than that in the other three parts. The reason may be that the roots were in contact with the soil for a long time, resulting in the microbial species in the soil being extremely rich.

The crude lipid extracts were fractionated via TLC, and the typical TLC results of some crude lipids and standard compounds are shown in Figure 1. The yields of CDW and crude lipid extracts from endophytic fungal isolates after cultivation at 28 °C for 7 days in PDB medium are given in Appendix A. The productivities of TAG and DAG for each endophytic fungal isolate were qualitatively estimated according to the brightness of the corresponding TLC bands, and the results are summarized in Appendix A. It is noted in Appendix A that only 5 of the 38 strains had biomass yields above 10.0 g/L, and only 12 strains had lipid yields higher than 1.0 g/L. It can be also seen from Appendix A that only several strains of endophytic fungi could accumulate relatively large amounts of DAG. After a final comprehensive evaluation, MLP41, MLG23, MLY23, MLY31W, and MLGP11 were selected as the DAG-producing strains and were subjected to further biological identification and fermentation for DAG production.

### 3.2. Identification of DAG-Producing Endophytic Fungi

The five selected isolates coded as MLP41, MLG23, MLY23, MLY31W, and MLGP11 were identified based on morphological features and ITS sequences. Their morphological features under a microscope are indicated in Figure 2. The morphological characteristics of the isolate coded as MLP41 showed sickle-shaped or oval spores with a diaphragm (Figure 2a). The conidia of MLG23 performed in solitary or clustered spores, and the hyphae were slender and septate (Figure 2b). The conidia of MLY23 exhibited short sticks and almost uniform size (Figure 2c). For MLY31W, its mycelium had a septum, and no spores were produced (Figure 2d). The morphological characteristics of MLGP11 included broom-like structures (Figure 2e). The morphological characteristics of MLP41, MLG23, MLY23, MLY31W, and MLGP11 under a microscope were consistent with those of the corresponding genera *Fusarium* sp., *Trichoderma* sp., *Colletotrichum* sp., *Pestalotiopsis* sp., and *Penicillium* sp., respectively.

The agarose gel electrophoresis results of PCR products of DNA samples extracted from MLP41, MLG23, MLY23, MLY31W, and MLGP11 are shown in Figure 2f. The lengths of the ITS sequences of the five strains were between 500 and 750 bp.

The sequencing and splicing results are given in Appendix A. The sequencing results were assembled with Contig Express, and the parts with inaccurate ends were removed. The spliced sequences were then aligned in the NCBI database, and the species with the highest homology was selected. The biological identification results of the five selected endophytic fungi coded as MLGP11, MLP41, MLY31W, MLG23, and MLY23 are summarized in Table 1. The ITS sequence length of MLP41 was 531 bp, which showed 99.81% similarly with *Fusarium annulatum* (*F. annulatum*) strain. The ITS sequence length of MLG23 was 572 bp, which was 99.83% similar to *Trichoderma dorotheae* (*T. dorotheae*). The ITS sequence length of MLY23 was 535 bp, which was 100% similar to *Colletotrichum aeschynomenes* (*C. aeschynomenes*). The ITS sequence length of MLY31W was 575 bp, which was 99.65% similar to *Pestalotiopsis scoparia* (*P. scoparia*). The ITS sequence length of MLGP11 was 561 bp, which was 100% similar to *Penicillium cataractarum* (*P. cataractarum*). As shown in Appendix A, the phylogenetic analysis also supported the identification results given in Figure 2 and Table 1.

### 3.3. FA compositions of Lipids and Their TLC Fractions from DAG-Producing Endophytic Fungi

Figure 3 shows the representative GC-MS ion profiles of total lipids, DAG, and TAG from the endophytic fungus MLY23. The detailed FA profiles of the crude lipids from the five screened endophytic fungi and their DAG and TAG components obtained after TLC fractionation are summarized in Table 2. The FAs of the crude lipids before TLC fractionation and of the obtained TAG, 1,2-DAG, and 1,3-DAG after TLC separation were obviously dominated by C16 and C18. In addition to the main FAs of palmitic acid (C16:0), linoleic acid (C18:2), oleic acid (C18:1), and stearic acid (C18:0), small amounts of myristic acid (C14:0), pentadecanoic acid (C15:0), palmitoleic acid (C16:1), arachidic acid (C20:0), and behenic acid (C22:0) were also detected in most of the crude lipids and pure TAG. Heptadecanoic acid (C17:0) and heptadecenoic acid (C17:1) were detected in only small amounts in the crude lipid and TAG fractions from MLP41, MLG23, and MLGP11, accounting for less than 1.40% (Table 2). A small amount of linolenic acid (C18:3) was observed in the crude lipid and TAG fractions from MLY23 and MLG23. The FA compositions of the crude lipids and their TAG fractions were more complex than those of the corresponding DAG fractions. However, the FAs of C17:0, C17:1, C18:3, C20:0, and C22:0 were not detected in all fractions of 1,2-DAG and 1,3-DAG. The FA types of the extracted lipids without TLC separation were similar to those of the corresponding pure TAG. The FA types of 1,2-DAG were almost the same as those of 1,3-DAG. Other signals in Figure 3 may be due to contaminants introduced by the reagents.

The relative contents of C16 and C18 accounted for higher than 94.07% of the content of all crude lipids from the five endophytic fungi and their TLC fractions of TAG, 1,2-DAG, and 1,3-DAG (Table 2). Except for those of MLY23, the contents of unsaturated fatty acids (UFAs) of DAG of the other four strains reached 60%, and the highest one reached 68.98% of 1,2-DAG from MLY31W. These UFAs were mainly composed of C18:1 and C18:2. DAG with a high content of UFAs has important practical application value. C18:1 and C18:2 are essential nutrients for the human body and can be used for immune regulation, treatment, and the prevention of different types of diseases, such as cardiovascular or autoimmune diseases, metabolic disorders, skin damage, and cancer, and also play an important role in drug absorption [22]. It can also be seen from Table 2 that the UFA contents of 1,2-DAG and 1,3-DAG from the same endophytic fungi were almost similar, but varied much among different strains, from 42.07% (1,3-DAG from MLY23) to 68.98% (1,2-DAG from MLY31W).

### 3.4. ^1^H NMR Spectra of the Extracted Endophytic Fungal Lipids and Determination of DAG Yields

The ^1^H NMR spectra of standards containing TAG, 1,2-DAG, 1,3-DAG, and MAG and the typical ^1^H NMR spectra of lipids extracted from the five endophytic fungi coded as MLP41, MLG23, MLY23, MLY31W, and MLGP11 are given in Appendix A. The assignments of the ^1^H NMR signals are summarized in Table 3. These assignments were in agreement with previous reports [23,24,25]. The ^1^H NMR spectra of the lipids extracted from the five endophytic fungi showed characteristic signals for all glycerides and free FAs. The strong signals at about 0.84–0.92 ppm corresponded to **–**C**H**_3_ from acyl groups and FA. The signals attributed to the other **–**(C**H**_2_)_n_**–** groups in FAs/esters were concentrated around 1.30 ppm. The signal at about 1.61 ppm was assigned to the **–**C**H**_2_**–** group attached to the carbonyl group at the *β* position, while the signal at about 2.28–2.38 ppm was assigned to the **–**C**H**_2_**–** group attached to the carbonyl group at the *α* position. The signals at about 2.00–2.04 (**–**C**H**_2_CH=CH**–**), 2.76 (**–**CH=CHC**H**_2_CH=CH**–**), and 5.30–5.40 ppm (**–**C**H**=C**H–**) indicated the presence of a large amount of UFAs, which was consistent with the results of FA compositions summarized in Table 2. The signal of **–**C**H**_2_OH from the glyceryl group in 1-MAG was observed at about 3.61 and 3.69 ppm. The signals at about 3.73 (**–**C**H**_2_OH), 4.07 (**–**C**H**OH**–**), and 4.13 ppm (**–**C**H**_2_OCO**–**) indicated the presence of 1,2-DAG and 1,3-DAG components. The characteristic signals at these three places could be well separated from the other signals at the resonance frequency of 600 MHz, and thus, they could be used to estimate the content of DAG in the extracted microbial lipids. The signals at about 3.84 ppm and 3.94 ppm were attributed to the **–**C**H**_2_OH of glyceryl groups in 2-MAG and the **–**C**H**OH**–** of glyceryl groups in 1-MAG, respectively. These two signals were very weak, indicating that the MAG content in the microbial lipids was very low. The symmetrical multiplet signals at about 4.10–4.38 ppm corresponded to the **–**C**H**_2_OCO**–** (sn-1, sn-3) ester signal of TAG, but the symmetrical pattern here only indicated the presence of TAG. After magnification, it could be seen that the signals here were not completely symmetrical, which was due to the overlap of the signals of DAG, MAG, and TAG. The very weak signal at about 5.08 ppm may be due to **–**C**H**(OCOR’)**–** from the glyceryl group in 1,2-DAG, and the strong peak at about 5.26 ppm corresponded to **–**C**H**(OCOR’)**–** from the glyceryl group in TAG. In addition, a stronger signal appeared at 3.67 ppm of microbial lipids from some strains (MLG23 and MLGP11), which may be polar lipids similar to phospholipids [26]. The signals at 3.98 and 3.99 ppm could not be determined presently.

There is a proportional relationship between the area of the ^1^H NMR signal and the number of protons that generate it [24]. Therefore, this method can be used to quantify the molar percentages of TAG, DAG, MAG, and free FAs in lipid samples simply, quickly, and accurately [24]. Here, the integral value of the signal of TMS (chemical shift 0.00) was normalized to 1.00000, and the mole number (N) of corresponding glyceride and free FA could be calculated according to formulas (3)–(8):N_1-MAG_ = *C**A_3.91-3.95_(3)
N_2-MAG_ = (*C**A_3.82-3.84_)/4(4)
N_1,2-DAG_ = (*C**A_3.72-3.74_)/2(5)
N_1,3-DAG_ = *C**A_4.05-4.10_(6)
N_TAG_ = (*C**A_4.11-4.38_-2N_1-MAG_-2N_1,2-DAG_-4N_1,3-DAG_)/4(7)
N_FA_ = (*C**A_2.28-2.38_-6N_TAG_-4N_1,2-DAG_-4N_1,3-DAG_-2N_1-MAG_-2N_2-MAG_)/2(8)
where *C* was the proportionality constant relating the ^1^H NMR spectral signal areas and the number of protons that generated them, A was the area of the spectral signal involved, and A_3.91-3.95_, A_3.82-3.84_, A_3.72-3.74_, A_4.05-4.10_, A_4.11-4.38_, and A_2.28-2.38_ were the areas of the signals ranging from 3.91 to 3.95, 3.82 to 3.84, 3.72 to 3.74, 4.05 to 4.10, 4.11 to 4.38, and 2.28 to 2.38 ppm, respectively.

The mole percentages of MAG, DAG, TAG, and FA were calculated according to formulas (9)–(14):MAG_(mol%)_ = 100(N_1-MAG_ + N_2-MAG_)/(N_1-MAG_ + N_2-MAG_ + N_1,2-DAG_ + N_1,3-DAG_ + N_TAG_ + N_FA_)(9)
1,2-DAG_(mol%)_ = 100N_1,2-DAG_/(N_1-MAG_ + N_2-MAG_ + N_1,2-DAG_ + N_1,3-DAG_ + N_TAG_ + N_FA_)(10)
1,3-DAG_(mol%)_ = 100N_1,3-DAG_/(N_1-MAG_ + N_2-MAG_ + N_1,2-DAG_ + N_1,3-DAG_ + N_TAG_ + N_FA_)(11)
DAG_(mol%)_ = 1,2-DAG_(mol%)_ + 1,3-DAG_(mol%)_(12)
TAG_(mol%)_ = 100N_TAG_/(N_1-MAG_ + N_2-MAG_ + N_1,2-DAG_ + N_1,3-DAG_ + N_TAG_ + N_FA_)(13)
FA_(mol%)_ = 100N_FA_/(N_1-MAG_ + N_2-MAG_ + N_1,2-DAG_ + N_1,3-DAG_ + N_TAG_ + N_FA_)(14)

It can be seen from the results of FA composition analysis that the FA compositions of these five endophytic fungal microbial lipids were mainly composed of C16 and C18 (>94%) (Table 2). According to the detailed FA compositions obtained via GC-MS, the average FA chain length of these five endophytic fungal microbial lipids was calculated as 17.5, and the C18 component accounted for 65.61–77.44% of the total FA composition. In order to facilitate the calculation, the C18 fatty acid with one double bond (oleic acid) was used as a reference to calculate the mass percentage of each glyceride, and the calculation was carried out according to formulas (15) and (16):1,2-DAG_(wt%)_ = (1,2-DAG_(mol%)_*M_DAG_)/(MAG_(mol%)_*M_MAG_ + DAG_(mol%)_*M_DAG_ + TAG_(mol%)_*M_TAG_ + FA_(mol%)_*M_FA_)(15)
1,3-DAG_(wt%)_ = (1,3-DAG_(mol%)_*M_DAG_)/(MAG_(mol%)_*M_MAG_ + DAG_(mol%)_*M_DAG_ + TAG_(mol%)_*M_TAG_ + FA_(mol%)_*M_FA_)(16)
where M_MAG_, M_DAG_, M_TAG_, and M_FA_ represented the average molecular weights of MAG, DAG, TAG, and free FA, respectively. Taking the average alkyl chain length as C18, then M_MAG_ = 356.54, M_DAG_ = 620.98, M_TAG_ = 885.43, and M_FA_ = 282.46.

Finally, the yields of 1,2-DAG, 1,3-DAG, and total DAG could be calculated according to formulas (17)–(19):(17)$$1,2-\mathrm{DAG}\;\mathrm{yield}\;(\mathrm{mg}/L)=\frac{\mathrm{Lipid}\;\mathrm{yield}\;\left(g/L\right)\times 1,2-\mathrm{DAG}\;\left(\mathrm{wt}\%\right)\times 1000}{100}$$
(18)$$1,3-\mathrm{DAG}\;\mathrm{yield}\;(\mathrm{mg}/L)=\frac{\mathrm{Lipid}\;\mathrm{yield}\;\left(g/L\right)\times 1,3-\mathrm{DAG}\;\left(\mathrm{wt}\%\right)\times 1000}{100}$$
Total DAG yield (mg/L) = 1,2-DAG yield + 1,3-DAG yield(19)

### 3.5. Effect of Fermentation Parameters on DAG Productivity

The five screened endophytic fungi coded as MLP41, MLG23, MLY23, MLY31W, and MLGP11 were cultured in K&R nitrogen-limited medium, and the effect of their fermentation parameters on the production of DAG were evaluated using the yields of lipids and DAG as the evaluation index.

#### 3.5.1. Effect of Fermentation Time

The maximum lipid and DAG accumulation, which varies in different oleaginous microorganisms, are greatly influenced by the fermentation time. Here, the yields of lipids and DAG after culturing for 3, 5, 7, 9, and 11 days are shown in Figure 4a–e. The other fermentation conditions were an inoculation dosage of 2 pieces/100 mL, temperature of 28 °C, and rotating speed of 120 rpm/min. Overall, the yields of lipids of the five endophytic fungi gradually increased with the extension of fermentation time. Due to the difference in the metabolic process, the fermentation time required to reach maximum yields of lipids and DAG was also different for different strains. In addition, the DAG yield varied with the lipid yield, indicating that DAG accumulation was closely related to lipid accumulation. However, the total yield of DAG (1,2-DAG + 1,3-DAG) did not always increase with the increase in the lipid yield. After 5 days of culture, the total DAG yields of MLP41, MLG23, and MLY23 reached the highest values of 24.67, 40.86, and 68.23 mg/L, respectively (Figure 4a–c). The maximum total DAG yield of 75.70 mg/L was obtained from MLY31W after 7 days of fermentation with a total lipid yield of 1.34 g/L (Figure 4d). Meanwhile, the highest total DAG yield of MLGP11 was obtained after 9 days of culture, reaching 71.58 mg/L (Figure 4e). The results indicated that all these five fungal strains had the potential to accumulate DAG.

From Figure 4a–e, it could be seen that the total yield of DAG reached the maximum value before that of lipids (except for MLGP11). Since the synthesis of lipids in oleaginous microorganisms is carried out in the order of FA synthesis, DAG synthesis, and then TAG synthesis [27,28], it was speculated that a higher DAG yield might be obtained at the early stage of lipid synthesis. In addition, the yield changes of 1,2-DAG and 1,3-DAG were basically synchronous, indicating that the accumulation of 1,2-DAG and 1,3-DAG in endophytic fungal cells was possibly in a state of dynamic equilibrium during fermentation. From Figure 4a–e, it could be also seen that 1,3-DAG was present in microbial lipids even at the early stage of fermentation for 3 or 5 days, indicating that the synthesis and decomposition of lipids in endophytic fungal cells might be performed simultaneously. In other words, the synthesized TAG would be hydrolyzed into sn-1,2-DAG during fermentation, and sn-1,2-DAG was unstable, resulting in its conversion to sn-1,3-DAG [29].

#### 3.5.2. Effect of Inoculation Dosage

Theoretically, if the inoculation dosage is too low, the limited number of fungal cells in the fermentation medium will eventually lead to low yields of the target products. However, too high of an inoculation dosage will accelerate the aging and autolysis of cells due to the competition of cells for nutrients under nutrient-limited conditions, resulting in a decrease in the lipid yield. Thus, after culturing MLP41, MLG23, MLY23, MLY31W, and MLGP11 for 5, 5, 5, 7, and 9 days, respectively, the yields of lipids and DAG are shown in Figure 4f–j. The other fermentation conditions were a temperature of 28 °C and rotating speed of 120 rpm/min. It can be seen from Figure 4f–j that the inoculation dosage had a significant impact on lipid production and the DAG yield. Generally speaking, with an increase in the inoculation amount, the lipid yield and DAG yield of the five endophytic fungi all showed a trend of first increasing and then decreasing. The maximum total DAG yields of the five endophyte fungi were obtained when the inoculation amount was 3 pieces per 100 mL of medium. The highest DAG yields of MLP41, MLG23, MLY23, MLY31W, and MLGP11 reached 113.19, 125.63, 114.99, 95.36, and 76.64 mg/L, respectively.

#### 3.5.3. Effect of Temperature

Temperature plays a very critical role in microbial growth and lipid accumulation. Here, the five screened isolates were cultured at 18, 23, 28, 33, and 38 °C for their corresponding optimized fermentation time with an inoculation dosage of 3 pieces/100 mL of medium at the rotating speed 120 rpm/min of flasks, and the yields of lipids and DAG are given in Figure 4k–o. The speed of lipid accumulation varied at different temperatures. The effect of temperature on the lipid accumulation rate of the MLP41 strain was different from that of the other four strains. As the temperature increased from 18 to 38 °C, the lipid yield of MLP41 decreased from 1.28 to 0.47 g/L (Figure 4k). However, the total DAG yield of MLP41 reached the maximum value of 113.19 mg/L at 28 °C (Figure 4k). For the MLG23 strain, its lipid yield showed the maximum value at 23 °C, but the maximum total DAG yield of 125.63 mg/L with a lipid yield of 1.01 g/L was obtained at 28 °C (Figure 4l). The yields of DAG from MLP41 and MLG23 reached their maximum values at different temperatures, indicating that their DAG accumulation occurred at different optimum temperatures. This may be due to differences in the optimal temperature of enzymes required for growth of microbial cell and production of secondary metabolites. The isolates coded as MLY23, MLY31W, and MLGP11 obtained the maximum lipid yield and DAG yield at 28 °C, and the maximum total DAG yields were 114.99, 95.36, and 76.64 mg/L, respectively.

#### 3.5.4. Effect of Rotating Speed of Flasks

The synthesis of microbial lipids is an aerobic process. Usually, the solubility of oxygen in the fermentation broth is low, and thus, the oxygen supply is an important limiting factor in the aerobic fermentation process [30]. Oxygen limitation may affect microbial growth and lipid accumulation. Therefore, in this study, after fermentation for their corresponding optimized times at 28 °C, with an inoculation dosage of 3 pieces/100 mL of medium and different rotating speeds of flasks, the yields of lipids and DAG with the five screened endophytic fungi were analyzed, and the results are shown in Figure 4p–t. Except for those of strain MLG23, the lipid yields of the other four strains increased gradually with an acceleration of the rotating speed of the shaken flasks. Yen and Zhang [31] found that low levels of dissolved oxygen promote lipid accumulation in the oleaginous yeast *Rhodotorula glutinis* while inhibiting the growth of its cells. On the other hand, high levels of dissolved oxygen could promote biomass enhancement rather than lipid accumulation [31]. Among the five strains, only the strain MLG23 followed this pattern obtained by Yen and Zhang [31]. This may be related to the different growth characteristics of different strains themselves.

The glycolysis cycle and tricarboxylic acid cycle process, involved in microbial lipid synthesis, is an aerobic process [32]. Glucose produces NADH and FADH_2_ through these two processes, which requires the transfer of H^+^ to O_2_ through the electronic respiratory chain and generates a large amount of ATP [32]. Some reactions of the glycolysis process require intervention with ATP [32]. Under hypoxic conditions, NADH and FADH_2_ cannot be completely converted into ATP in microbial cells, and thus, the ATP supplied to the glycolysis pathway is reduced, leading to a decrease in the rate of glucose conversion and lipid synthesis [32]. Therefore, in the present experiment the low yields of lipids were obtained at a low rotating speed of the flasks. It is also seen from Figure 4p–t that too low or too high of a shake flask rotating speed was not conducive to the accumulation of DAG. All five fungi obtained the highest total DAG yields at the rotating speed of 120 rpm. At this speed of 120 rpm, the fungi neither slowed down lipid synthesis due to a lack of oxygen nor exhibited difficulties in accumulating DAG due to excessive dissolved oxygen.

#### 3.5.5. Effect of C/N Ratio

An important biochemical regulatory mechanism of lipid accumulation is nitrogen limitation [33]. When oleaginous microorganisms grow under nitrogen-deficient conditions (high C/N ratio), the excess carbon source will stimulate the synthesis of FAs through the de novo FA synthesis pathway and store them in the cells in the form of lipid droplets. The yields of lipids and DAG in the five strains cultured at different C/N ratios in medium are shown in Figure 4u–y. The other culture conditions were an inoculation dosage of 3 pieces/100 mL and a flask rotating speed of 120 rpm/min at 28 °C for the optimal time. It is obvious that the C/N ratio greatly affected the yields of lipids and DAG. For the fungal strains MLP41 and MLG23, the highest total DAG yields of 112.28 and 126.42 mg/L were obtained with a C/N ratio of 38 (g/g), and further increasing the C/N ratio had no significant effect on improving the yield of DAG. The change in the C/N ratio had little effect on the DAG yield of MLY31W, and the highest total DAG yield was only 105.61 mg/mL at a C/N ratio of 98 (g/g). The total DAG yields of MLY23 and MLGP11 reached the highest values of 189.87 and 135.56 mg/L at the C/N ratio of 68 (g/g). MLY23 obtained the highest total DAG yield among the five endophytic fungi, reaching 189.87 mg/L with 1,2-DAG of 77.55 mg/mL and 1,3-DAG of 112.32 mg/mL. The results suggested that the appropriate C/N ratio would be helpful for DAG production in some fungal strains.

The above results indicated that the yields of 1,3-DGA were much higher than those of 1,2-DGA under the corresponding fermentation conditions of each strain (Figure 4). In addition, the highest DAG yields of the five screened endophytic fungi were much higher than those in the literature [10,11]. The high content of 1,3-DAG in microbial lipids makes it more advantageous for utilization in functional foods, which is beneficial to reduce fat accumulation in the blood. It is suggested that the lipids obtained from the five endophytic fungal isolates have great potential for development in functional foods with a hypolipidemic effect. It is also indicated that the fungus *C. aeschynomenes* coded as MLY23 has the greatest potential for DAG production.

## 4. Conclusions

Five endophytic fungal strains from *T. chinensis* were screened for DAG production. They belonged to different genera: *F. annulatum* (coded as MLP41), *T. dorotheae* (coded as MLG23), *C. aeschynomenes* (coded as MLY23), *P. scoparia* (coded as MLY31W), and *P. cataractarum* (coded as MLGP11). The crude lipids from the five strains and their corresponding TAG, 1,2-DAG, and 1,3-DAG, fractionated via TLC, were mainly composed of C16:0, C18:2, C18:1, and C18:0, which in total, accounted for higher than 94% (C16 + C18). *C. aeschynomenes* had the strongest DAG production ability, and the highest total DAG yield reached 189.87 mg/mL. The results showed that it may be a very interesting route to produce DAG via specific endophytic fungal fermentation, such as with *C. aeschynomenes*. Since DAG is the last key intermediate in the synthesis of TAG in microbial cells, it is meaningful to further improve the DAG yield through gene engineering, enzyme engineering, and other molecular biological methods.

## Figures and Tables

**Figure 1 foods-12-00399-f001:**
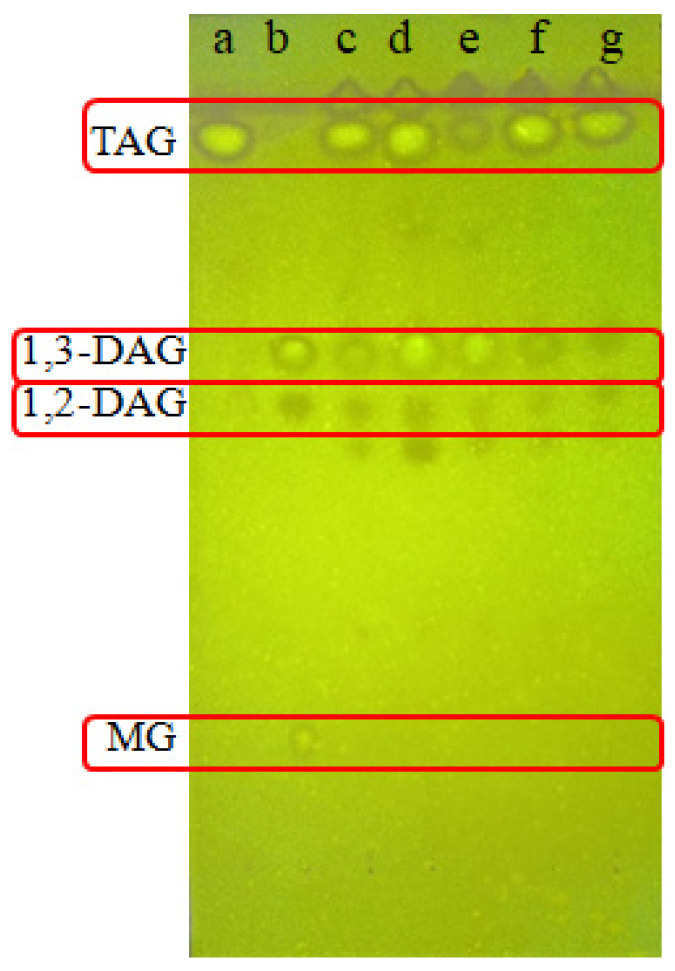
TLC bands of (**a**) TAG standard, (**b**) DAG standard (containing small amounts of MAG), (**c**) lipids from MLY23, (**d**) lipids from MLG23, (**e**) lipids from MLGP11, (**f**) lipids from MLP41, and (**g**) lipids from MLY31W.

**Figure 2 foods-12-00399-f002:**
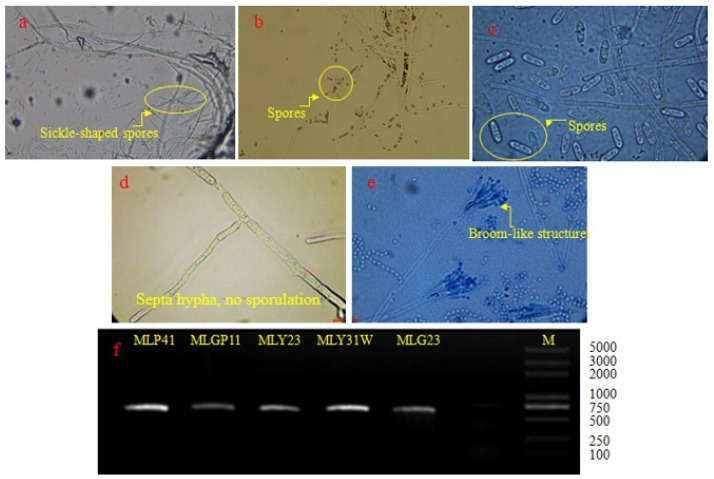
Morphological features of (**a**) MLP41, (**b**) MLG23, (**c**) MLY23, (**d**) MLY31W, and (**e**) MLGP11 under a microscope (1000× magnification), and (**f**) agarose gel electrophoresis of PCR products.

**Figure 3 foods-12-00399-f003:**
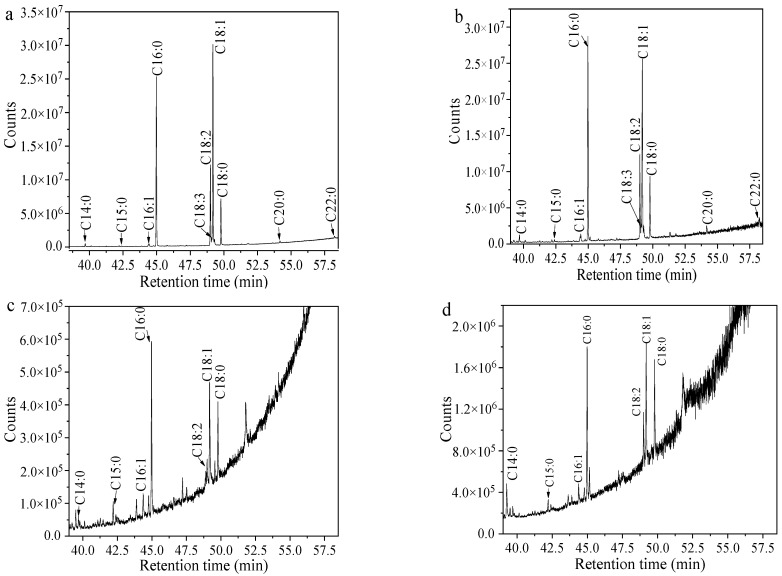
Typical GC-MS ion chromatogram of lipids from the endophytic fungus MLY23 and its DAG and TAG components: (**a**) Lipids before TLC separation, (**b**) TAG, (**c**) 1,2-DAG, (**d**) 1,3-DAG.

**Figure 4 foods-12-00399-f004:**
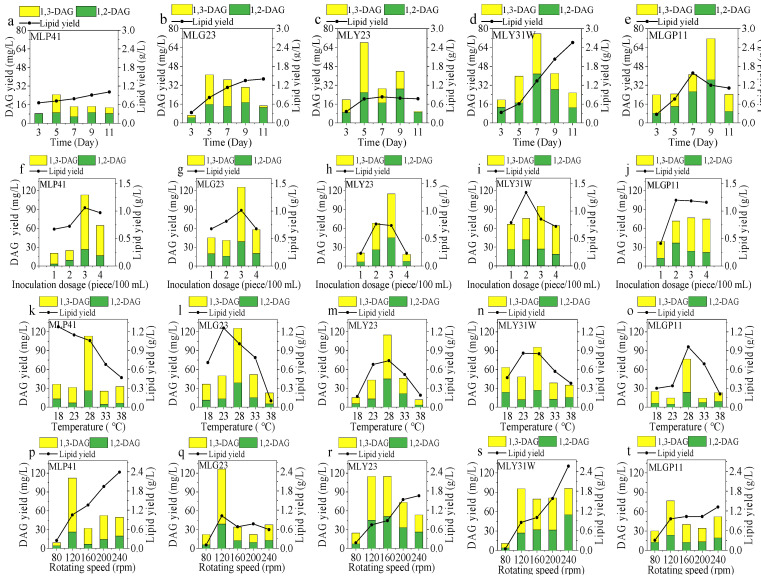
Effect of fermentation conditions on the yields of lipids and DAG from the five fungi coded as MLP41, MLG23, MLY23, MLY31W, and MLGP11: (**a**–**e**) fermentation time, (**f**–**j**) inoculation dosage, (**k**–**o**) fermentation temperature, (**p**–**t**) rotating speed of flasks, and (**u**–**y**) C/N ratio.

**Table 1 foods-12-00399-t001:** Biological identification results of the five selected endophytic fungi.

Code	Tissue	Identification Result	Accession Number	Homology
MLP41	Bark	*Fusarium annulatum* (*F. annulatum*)	OK325613	559/559 (100%)
MLG23	Root	*Trichoderma dorotheae* (*T. dorotheae*)	OK315570	535/535 (100%)
MLY23	Leaf	*Colletotrichum aeschynomenes* (*C. aeschynomenes*)	OK325610	530/531 (99.81%)
MLY31W	Leaf	*Pestalotiopsis scoparia* (*P. scoparia*)	OK325612	573/575 (99.65%)
MLGP11	Root	*Penicillium cataractarum* (*P. cataractarum*)	OK325611	571/572 (99.83%)

**Table 2 foods-12-00399-t002:** FA compositions of crude lipids and their DAG and TAG components from the five screened endophytic fungi.

FAs	Relative Percentage (%)																			
	MLP41				MLG23				MLY23				MLY31W				MLGP11			
	Lipids	TAG	1,2-DAG	1,3-DAG	Lipids	TAG	1,2-DAG	1,3-DAG	Lipids	TAG	1,2-DAG	1,3-DAG	Lipids	TAG	1,2-DAG	1,3-DAG	Lipids	TAG	1,2-DAG	1,3-DAG
C14:0	0.41	0.36	1.12	nd	0.003	0.51	nd	nd	0.56	1.25	3.08	1.74	0.42	nd	nd	0.49	0.54	0.18	nd	nd
C15:0	0.31	0.53	nd	nd	0.43	0.81	0.64	0.43	nd	0.37	1.70	1.46	nd	nd	nd	nd	1.73	1.04	0.98	nd
C16:0	21.58	28.45	20.24	24.82	19.08	25.21	27.82	25.97	31.97	31.61	35.11	34.13	29.09	34.24	21.80	19.15	17.63	24.55	23.33	24.43
C16:1	1.14	2.04	2.38	nd	0.77	1.93	2.30	1.46	0.85	2.10	7.22	5.43	0.44	nd	nd	1.62	1.44	0.77	2.11	2.01
C17:0	0.08	nd	nd	nd	0.40	0.52	nd	nd	nd	nd	nd	nd	nd	nd	nd	nd	1.25	0.33	nd	nd
C17:1	nd	nd	nd	nd	0.59	1.37	nd	nd	nd	nd	nd	nd	nd	nd	nd	nd	0.70	nd	nd	nd
C18:0	10.93	10.07	11.56	15.36	7.15	8.07	7.03	8.71	9.15	10.93	16.31	20.61	10.18	11.26	9.22	12.69	13.88	17.42	14.18	14.59
C18:1	38.66	34.20	40.56	46.69	36.76	32.55	31.12	34.99	40.41	33.55	31.34	26.77	38.61	38.92	45.69	55.82	26.82	26.81	34.00	28.80
C18:2	25.81	24.04	24.14	13.13	32.12	28.96	31.09	28.43	14.71	14.83	5.24	9.87	19.52	12.89	23.29	10.23	34.30	27.65	25.40	30.16
C18:3	nd	nd	nd	nd	1.41	nd	nd	nd	1.34	2.20	nd	nd	nd	nd	nd	nd	nd	nd	nd	nd
C20:0	0.46	0.09	nd	nd	0.32	0.08	nd	nd	0.46	1.47	nd	nd	1.16	2.09	nd	nd	1.72	0.53	nd	nd
C22:0	0.61	0.21	nd	nd	nd	nd	nd	nd	0.55	1.69	nd	nd	0.58	0.60	nd	nd	nd	0.71	nd	nd
C16 + C18	98.03	98.81	98.88	100.00	97.29	96.71	99.36	99.57	98.43	95.22	95.22	96.80	97.84	97.31	100.00	99.51	94.07	94.42	99.02	100.00
SFA	34.39	39.72	32.92	40.18	27.38	35.19	35.49	35.12	42.69	47.32	56.92	57.93	41.43	48.19	31.02	32.33	36.74	46.35	38.49	39.03
MUFA	39.80	36.24	42.94	46.69	39.09	35.85	33.42	36.45	41.26	35.65	38.56	32.20	39.05	38.92	45.69	57.44	28.95	26.79	36.11	30.81
PUFA	25.81	24.04	24.14	13.13	33.53	28.96	31.09	28.43	16.05	17.03	5.24	9.87	19.52	12.89	23.29	10.23	34.30	26.86	25.40	30.16
UFA	65.61	60.28	67.08	59.82	72.62	64.81	64.51	64.88	57.31	52.68	43.80	42.07	58.57	51.81	68.98	67.67	63.26	53.65	61.51	60.97

Lipids, crude lipids without TLC fractionation; nd, not detected; SFA, saturated fatty acid; MUFA, monounsaturated fatty acid; PUFA, polyunsaturated fatty acid; UFA, unsaturated fatty acid.

**Table 3 foods-12-00399-t003:** Chemical shift assignments and multiplicities of the ^1^H NMR signals of the main protons of glycerides and FAs present in lipids.

Code	Chemical Shift (ppm)	Multiplicity	Type of Protons	Compound
A	0.84–0.92	t	**–**C**H**_3_	Acyl groups and FA
B	1.30	m	**–**(C**H**_2_)_n_**–**	Acyl groups and FA
C	1.61	m	**–**OCOCH_2_C**H**_2_**–**	Acyl groups
D	2.00–2.04	m	**–**C**H**_2_CH=CH**–**	Acyl groups and FA
E	2.28–2.38	m	**–**OCOC**H**_2_**–**	Acyl groups and FA
F	2.76	m	**–**CH=CHC**H**_2_CH=CH**–**	Unsaturated *ω*-6 and *ω*-3 acyl groups and FA
G	3.61, 3.69	dd	**–**C**H**_2_OH	Glyceryl group in 1-MAG
H	3.73	m	**–**C**H**_2_OH	Glyceryl group in 1,2-DAG
I	3.84	m	**–**C**H**_2_OH	Glyceryl group in 2-MAG
J	3.94	m	**–**C**H**OHCH_2_OH	Glyceryl group in 1-MAG
K	4.07	m	**–**C**H**OH**–**	Glyceryl group in 1,3-DAG
L	4.13	dd	**–**C**H**_2_OCO**–**	Glyceryl group in 1,3-DAG
M	4.14	dd	**–**C**H**_2_OCO**–**	Glyceryl group in TAG
N	4.18	dd	**–**C**H**_2_OCO**–**	Glyceryl group in 1-MAG
O	4.23	dd	**–**C**H**_2_OCO**–**	Glyceryl group in 1,2-DAG
P	4.29	dd, dd	**–**C**H**_2_OCO**–**	Glyceryl group in TAG
Q	5.08	m	**–**C**H**(OCOR’)**–**	Glyceryl group in 1,2-DAG
R	5.26	m	**–**C**H**(OCOR’)**–**	Glyceryl group in TAG
S	5.30–5.40	m	**–**C**H**=C**H****–**	Acyl groups and FA
T	3.99	dd		Undetermined
U	3.98	dd		Undetermined
V	3.67	m		Phospholipids

Abbreviations: t, triplet; m, multiplet; d, double.

## Data Availability

The datasets generated for this study are available on request to the corresponding author.

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
