# Peer review of "Fermentative Production of Diacylglycerol by Endophytic Fungi Screened from Taxus chinensis var. mairei"

_foods, 2023, doi:10.3390/foods12020399_

Round 1

Reviewer 1 Report

In general, the authors present lots of data and in my opinion. My suggestions are listed as follows.

1)In the abstract, line 15-16, the full names should be provided for F. annulatum, T. dorotheae, C. aeschynomenes, P. scoparia, and P. cataractarum in their first presence.

2) the manuscript should be improved. For example, Figure 1, 2 look not normal.

3) the authors pointed out many times that they optimized the fermentation conditions. But what we see in the manuscript, the authors just analyzed the single-factor effect. Usually, they should further optimize based on orthogonal experiments or response surface analysis.

Author Response

All the revisions have been highlighted in green color.

Reviewer 1#

In general, the authors present lots of data and in my opinion. My suggestions are listed as follows.

Point 1: In the abstract, line 15-16, the full names should be provided for F. annulatum, T. dorotheae, C. aeschynomenes, P. scoparia, and P. cataractarum in their first presence.

Response 1: In the abstract, the full names have been provided for F. annulatum, T. dorotheae, C. aeschynomenes, P. scoparia, and P. cataractarum in their first presence. The full names are Fusarium annulatum,Trichoderma dorotheae, Colletotrichum aeschynomenes, Pestalotiopsis scoparia, and Penicillium cataractarum, respectively.

Point 2: the manuscript should be improved. For example, Figure 1, 2 look not normal.

Response 2: We checked the manuscript carefully again. Figure 1 was the photo of original TLC plate and it was taken by the camera. Figure 2 was also the original images. Now we have edited Figure 1 and Figure 2 again, making them looking more normal.

Point 3: the authors pointed out many times that they optimized the fermentation conditions. But what we see in the manuscript, the authors just analyzed the single-factor effect. Usually, they should further optimize based on orthogonal experiments or response surface analysis.

Response 3: After deep consideration, the expressions about optimization of the fermentation conditions are not suitable. Here, only the single-factor effects on the lipid accumulation and DAG productivity, including cultivation time, inoculation dosage, temperature, rotating speed of flasks (dissolved oxygen), and the mass ratio of carbon to nitrogen (C/N ratio), were discussed. We discussed the next effecting factor based on the results of the previous one. For example, when we evaluated the effect of inoculation dosage, the former optimal fermentation time was adopted in the experiments. So the relative expressions about optimization of the fermentation conditions are revised. For example:

In the abstract, “The fermentation conditions were optimized,” is revised to “The effects of fermentation conditionson the DAG productivitywere discussed,”.

“2.5 Optimization of culture conditions to maximize DAG productivity” is revised to “2.5 Evaluation of culture conditions on DAG productivity”.

Line 89, “Finally, the fermentation conditions were optimized.” is revised to “Finally, the effects of fermentation conditionson the DAG productivity were analyzed.”

Line 101, “3.5 Optimization of fermentation parameters” is revised to “3.5 Effect of fermentation parameterson DAG productivity”.

Line 249-251, “C. aeschynomenes performed the strongest ability for DAG production, and the total DAG yield reached 189.87 mg/mL under the optimal fermentation conditions.” is revised to “C. aeschynomenes performed the strongest ability for DAG production, and the highest total DAG yield reached 189.87 mg/mL.”

Reviewer 2 Report

Dear authors.

The problems of obesity are peculiar to the modern world. Moreover, hunger and obesity are closely related to each other and to the quality of food consumed by people. However, obesity is not provoked by fat or any other components of natural products. Trans fats and hidden sugars are found in large quantities in industrial products. They are the main culprits of obesity. The scientific content of the manuscript justifies its publication in scientific journal. Some additions and modifications will significantly improve the quality of the article, but it not suitable for Foods.

Major comments:

1) It is better to correct the title of the article.

2) Abstract, the purpose and prospects of using the results should be added.

3) Introduction, the goal should be formulated.

4) The introduction does not justify the presentation of this material in Foods.

5) M&M, the designation m2 in formula (2) is not clear. This parameter is related to the formula (1)?

6) M&M, Which methods allowed optimization of the yeast cultivation process?

7) L.213, 329. Adjust the section number.

8) Table 2, nd – requires an explanation.

9) The discussion is presented in insufficient volume.

10) Conclusions, prospects for using the results obtained should be added.

11) In the References, 31% of publications refer to 2018-2022 (the last 5 years); the remaining 69% of used sources are older than 5 years. It is recommended to increase the share of references to sources published over the last 5 years when analyzing the current state of research in the area under consideration, since this area of knowledge is rapidly developing.

Author Response

All the revisions have been highlighted in green color.

Reviewer 2#

Major comments:

Point 1: It is better to correct the title of the article.

Response 1: After careful consideration, the title of the article “Production of diacylglycerol by endophytic fungal fermentation: Screening and identification of endophytic fungi and optimization of fermentation conditions” is changed to “Fermentative production of diacylglycerol by endophytic fungal screened from Taxus chinensis var. mairei”.

Point 2: Abstract, the purpose and prospects of using the results should be added.

Response 2: In the abstract, the purpose and prospects of using the results has been added as “The results showed that it may be a new promising route for the production of DAG by specific endophytic fungal fermentation, such as C. aeschynomenes.”

Point 3: Introduction, the goal should be formulated.

Response 3: The introduction has been revised carefully, making the goal is much more formulated. The goal of the present study was to obtain one or more endophytic fungi strains which had a great potential for DAG production.

Point 4: The introduction does not justify the presentation of this material in Foods.

Response 4: The introduction has been revised carefully. Now the content is more logical, making the research goal be more clear.

DAG is an important structural lipid and plays an important role in human health, which has been widely recognized. DAG is also an important additive of many foods, which has been widely used in food industry. In the second paragraph of Introduction, the presentation of DAG in foods has been stated.

Point 5: the designation m2 in formula (2) is not clear. This parameter is related to the formula (1)?

Response 5: The designation m2 in formula (2) is the productivity of CDW (g/L). This parameter is related to m2 in the formula (1). We have revised the expression of the designation m2 in formula (2) to “where m2 was the productivity of CDW (g/L)”.

Point 6: Which methods allowed optimization of the yeast cultivation process?

Response 6: After deep consideration, the expressions about optimization of the fermentation conditions are not suitable. Here, only the single-factor effects on the lipid accumulation and DAG productivity, including cultivation time, inoculation dosage, temperature, rotating speed of flasks (dissolved oxygen), and the mass ratio of carbon to nitrogen (C/N ratio), were discussed. We discussed the next effecting factor based on the results of the previous one or ones. For example, when we evaluated the effect of inoculation dosage, the former optimal fermentation time was adopted in the experiments. Based on the single-factor effects on the lipid accumulation and DAG productivity, the cultivation rules of five fungi can be obtained. So the relative expressions about optimization of the fermentation conditions are revised. For example:

In the abstract, “The fermentation conditions were optimized,” is revised to “The effects of fermentation conditionson the DAG productivitywere discussed,”.

“2.5 Optimization of culture conditions to maximize DAG productivity” is revised to “2.5 Evaluation of culture conditions on DAG productivity”.

Line 89, “Finally, the fermentation conditions were optimized.” is revised to “Finally, the effects of fermentation conditionson the DAG productivity were analyzed.”

Line 101, “3.5 Optimization of fermentation parameters” is revised to “3.5 Effect of fermentation parameterson DAG productivity”.

Line 249-251, “C. aeschynomenes performed the strongest ability for DAG production, and the total DAG yield reached 189.87 mg/mL under the optimal fermentation conditions.” is revised to “C. aeschynomenes performed the strongest ability for DAG production, and the highest total DAG yield reached 189.87 mg/mL.”

Point 7: L.213, 329. Adjust the section number.

Response 7: In our submitted manuscript, in Line 213, the section number was “2.6. 1H NMR analysis of the extracted lipids and determination of DAG yield”. It was right. However, in the downloaded manuscript from the submission system, which has been edited by the editors, the section number was changed to “2.61. H NMR analysis of the extracted lipids and determination of DAG yield”. Now we have corrected it back.

In our submitted manuscript, in Line 329, the section number was “3.4. 1H NMR analysis of the extracted lipids and determination of DAG yield”. It was right. However, in the downloaded manuscript from the submission system, which has been edited by the editors, the section number was changed to “3.41. H NMR spectra of the extracted endophytic fungal lipids and determination of DAG yields”. Now we have corrected it back.

Point 8: Table 2, nd – requires an explanation.

Response 8: “nd” referred to “not detected”. The corresponding explanation has been added under Table 2 as a not.

Point 9: The discussion is presented in insufficient volume.

Response 9: We have checked and revised the full text once again. After deep consideration, the discussion of the results in the present paper is sufficient. If we just want to increase the volume of the discussion and add ineffective discussions, the whole article will look incongruous.

Point 10: Conclusions, prospects for using the results obtained should be added.

Response 10: In Conclusions, prospects for using the results obtained have been added. The results showed that it may be a very interesting route to produce DAG by specific endophytic fungal fermentation, such as C.aeschynomenes. Since DAG is the last key intermediate in the synthesis of TAG in microbial cell, it is meaningful to further improve the DAG yield through gene engineering, enzyme engineering, and other molecular biological methods.

Point 11: In the References, 31% of publications refer to 2018-2022 (the last 5 years); the remaining 69% of used sources are older than 5 years. It is recommended to increase the share of references to sources published over the last 5 years when analyzing the current state of research in the area under consideration, since this area of knowledge is rapidly developing.

Response 11: About 10 references have been renewed to those publicated between in 2019-2023. Now about 70% of publications refer to 2018-2023 (the last 5 years).

Reviewer 3 Report

Dear author,

Greetings!

1, Title

Production of diacylglycerol by endophytic fungal fermentation: Screening and identification of endophytic fungi and optimization of fermentation conditions

It is vague and so many and (please give appropriate title )

Abstract 

2Diacylglycerol (DAG) production by fermentation is meaningful. Five endophytic fungi 13 which could accumulate DAG were screened from Taxus chinensis var. mairei. The strains coded as 14 MLP41, MLG23, MLY23, MLY31W, and MLGP11 were biologically identified to be F. annulatum, T. 15 dorotheae, C. aeschynomenes, P. scoparia, and P. cataractarum,(Please revise )he highest total DAG yields of F. annulatum, T. dor-20 otheae, C. aeschynomenes, P. scoparia, and P. cataractarum were 112.28, 126.42, 189.87, 105.61, and 21 135.56 mg/L, respectively. C. aeschynomenes performed the strongest actually method or assay technique shold be highlighted/

Diacylglycerol is a compound found in small amounts in plant oils. Oils rich in diacylglycerol can be made in the lab and used to replace fats in the diet.

Diacylglycerol might work by increasing energy use and the breakdown of fat. Diacylglycerol-concentrated oils taste and look like regular fats and cooking oils. Please explain with fungus through kegg metabolic pathways

Author Response

All the revisions have been highlighted in green color.

Reviewer 3#

Point 1: Title

Production of diacylglycerol by endophytic fungal fermentation: Screening and identification of endophytic fungi and optimization of fermentation conditions

It is vague and so many and (please give appropriate title )

Response 1: After careful consideration, the title of the article “Production of diacylglycerol by endophytic fungal fermentation: Screening and identification of endophytic fungi and optimization of fermentation conditions” is changed to “Fermentative production of diacylglycerol by endophytic fungal screened from Taxus chinensis var. mairei”.

Point 2: Abstract

Diacylglycerol (DAG) production by fermentation is meaningful. Five endophytic fungi which could accumulate DAG were screened from Taxus chinensis var. mairei. The strains coded as MLP41, MLG23, MLY23, MLY31W, and MLGP11 were biologically identified to be F. annulatum, T. dorotheae, C. aeschynomenes, P. scoparia, and P. cataractarum, (Please revise) The highest total DAG yields of F. annulatum, T. dorotheae, C. aeschynomenes, P. scoparia, and P. cataractarum were 112.28, 126.42, 189.87, 105.61, and 21 135.56 mg/L, respectively. C. aeschynomenes performed the strongest actually method or assay technique shold be highlighted/

Diacylglycerol is a compound found in small amounts in plant oils. Oils rich in diacylglycerol can be made in the lab and used to replace fats in the diet.

Response 2: In the abstract, the full names have been provided for F. annulatum, T. dorotheae, C. aeschynomenes, P. scoparia, and P. cataractarum in their first presence. The full names are Fusarium annulatum,Trichoderma dorotheae, Colletotrichum aeschynomenes, Pestalotiopsis scoparia, and Penicillium cataractarum, respectively.

In the abstract, the actual technique has been highlighted as “The results showed that it may be a new promising route for the production of DAG by specific endophytic fungal fermentation, such as C. aeschynomenes.”

The introduction has been revised carefully, making the goal of the research is much more formulated. The goal of the present study was to obtain one or more endophytic fungi strains which had a great potential for DAG production.

Diacylglyceol is actually a compound found in small amounts in plants and animal fats. But it will be more significance and economic value if the production of DAG could be realized by fermentation.

Point 3: Diacylglycerol might work by increasing energy use and the breakdown of fat. Diacylglycerol-concentrated oils taste and look like regular fats and cooking oils. Please explain with fungus through kegg metabolic pathways

Response 3: Though diacylglycerol-concentrated oils taste and look like common fats and cooking oils. The function of diacylglycerol for human body is different from that of common oils and fats.

The main purpose of this study is to screen out the strains that can produce diacylglycerol, and analyze the influence of fermentation conditions on the production of diacylglycerol. In the future, when we theoretically study the pathway of producing diglycerides in fungal cells, it is necessary to use the kegg metabolic pathway to explain. This is another topic and a paper. Therefore, it is not helpful to use the kegg pathway to explain the metabolism of fungi in this study.

Round 2

Reviewer 2 Report

Dear Authors

My comments are taken into account

Author Response

The manuscript is checked and revised carefully again. All the changes have been highlighted in yellow.

Reviewer 3 Report

Dear authors,

Greetings!

Please modify abstract little bit with technical words and explain the type of fermentation ,screening procedure and media for enrichment  please change biologically identified and please remove promising route etc (please modify abstract)

2 Please check technical  english by using grammarly and mendeley

3please modify table -1 by inserting correct specimen of barks,roots and leaves

4 23,25 references please edit and also please check all references based on our journal format

5 Please include cotton blue mounting procedure and turn it in report for plagiarism

Best 

Author Response

Thank you very much for your suggestions. The manuscript is checked and revised carefully again. All the changes have been highlighted in yellow.

Point 1. Please modify abstract little bit with technical words and explain the type of fermentation, screening procedure and media for enrichment. Please change biologically identified and please remove promising route etc (please modify abstract).

Response:

The abstract is carefully modified. The type of fermentation, screening procedure, and media is explained in the abstract. In the present study, five endophytic fungi which could accumulate DAG were screened from Taxus chinensis var. mairei by using potato dextrose agar plate and flask cultivation in potato dextrose broth culture medium. The biological identification methods are added. The expression of identification results is revised.

After deep consideration, the promising route should not be removed. In the Round 1 review, one reviewer also suggested that the promising route should be supplied.

Point 2. Please check technical english by using grammarly and Mendeley.

Response: The manuscript is checked and revised carefully again. The changes have been highlighted in yellow.

Point 3. Please modify table -1 by inserting correct specimen of barks, roots and leaves.

Response:

The specimen of barks, roots and leaves in Table 1 has been modified to bark, root and leaf, respectively.

Point 4. 23,25 references please edit and also please check all references based on our journal format.

Response:

All references have been checked carefully based on the journal format of Foods.

Point 5. Please include cotton blue mounting procedure and turn it in report for plagiarism.

Response:

We do not understand what the cotton blue mounting procedure is. Here, we show you a report for plagiarism through Web of Science (https://www.webofscience.com). This report is also called as duplicate checking report. The search topics are diacylglycerol and endophytic fungi, the search result is shown as below. This search result shows that our article is not plagiarized from others.
